# What Should I Notice? Using Algorithmic Information Theory to Evaluate the Memorability of Events in Smart Homes

**DOI:** 10.3390/e24030346

**Published:** 2022-02-27

**Authors:** Étienne Houzé, Jean-Louis Dessalles, Ada Diaconescu, David Menga

**Affiliations:** 1SEQUOIA, EDF R&D, 7 Boulevard Gaspard Monge, 91120 Palaiseau, France; david.menga@edf.fr; 2INFRES, Télécom Paris, 19 Place Marguerite Perey, 91120 Palaiseau, France; jean-louis.dessalles@telecom-paris.fr (J.-L.D.); ada.diaconescu@telecom-paris.fr (A.D.)

**Keywords:** Kolmogorov complexity, algorithmic information theory, simplicity, abduction, memorability

## Abstract

With the increasing number of connected devices, complex systems such as smart homes record a multitude of events of various types, magnitude and characteristics. Current systems struggle to identify which events can be considered more memorable than others. In contrast, humans are able to quickly categorize some events as being more “memorable” than others. They do so without relying on knowledge of the system’s inner working or large previous datasets. Having this ability would allow the system to: (i) identify and summarize a situation to the user by presenting only memorable events; (ii) suggest the most memorable events as possible hypotheses in an abductive inference process. Our proposal is to use Algorithmic Information Theory to define a “memorability” score by retrieving events using predicative filters. We use smart-home examples to illustrate how our theoretical approach can be implemented in practice.

## 1. Introduction

Let us consider the following scenario. As a user has just switched on the TV for the first time in her new all-equipped living room, the lights dim and the window blinds go down. Intrigued by this behavior, she quickly infers that both light dimming and blind closing occurred as a consequence of the TV being switched on. How did she come to this conclusion? By performing *abductive inference* [1]. This mental operation is a key element of the human ability to understand the world: from theobservation, we infer the possible causes.

In this example, there are mainly three ways in which the user could come to the conclusion. (1) If the user knows how the smart living-room system works, if she knows the underlying rules or parameters, she may use this causal knowledge to perform abduction. (2) If she has no knowledge about the system but has made several observations of the same behavior, she may examine past correlations and figure out that turning on the TV set often leads the blinds to close and the lights to dim. (3) If there are no previous occurrences of the event (e.g., it is the first time she has switched on the TV in the living room), she may still be able to suspect that the TV is a possible cause for the observed event, just because it appears to her as a memorable recent event (as it is its first occurrence). This example suggests that human beings are able to use at least three distinct methods to perform abductive tasks and infer new knowledge, depending on the situation. While the first two mechanisms are commonly used in Explainable AI literature [2] to identify causes and explanations of agents’ decisions, they require preliminary knowledge or data. On the other hand, the third approach can be used without any previous knowledge of the occurring phenomenon or of its past occurrences but remains, to the best of our knowledge, not implemented in current systems. Doing so would require the system to have a way of distinguishing some events as more “memorable” than others and then consider them as possible hypotheses if necessary [1].

Defining a memorability score is not straightforward. First, events can be of different nature, and not directly comparable. For systems such as smart homes, noticeable events range from device removal to presence detection or unusually high temperatures. Even for comparable events, the problem is to weigh different characteristics: is a record-high temperature 47 days ago more memorable than the small deviation recorded just 3 minutes ago? To our knowledge, no current system proposes to combine various event types from different devices to compute a unified metric of “memorability”. In addition to the aforementionned use for abductive inference, having access to a computation of memorability would allow a system to summarize a situation to its user by presenting only the most memorable events: for instance, a summary of notable events that occurred during the home owner’s absence.

To address this issue, we started from the following supposition: while all events, regardless of their characteristics or nature, can be uniquely described using a combination of quantitative or qualitative qualifiers, the most memorable ones are likely to require fewer words to be described. This supposition appears to be in line with observations of human cognition: for instance, a correlation has been found between word frequency and length [3], the shortest words being the most common; moreover, humans seem to be sensitive to the complexity of events when assessing a coincidence [4,5]. To illustrate this, consider, for instance, the “182nd day of 7 years ago” compared to “the hottest day ever recorded”. Similarly, how does “the day of my wedding” compare to “the 10th of September 8 years ago”? In both cases, one occurrence seems more memorable than the other: how can we quantify this relative simplicity? We propose to evaluate the complexity of each description, taking into account both the complexity of the concept words (a date of occurrence, a temperature ranking), and of the arguments (“hottest” vs. 182 and 7). The resulting values define the *description complexity* of events. Following our supposition, we would define memorable events as requiring simpler and less numerous qualifiers to be unambiguously described than unremarkable ones.

For machines to implement and compute description complexity, we need a formal framework and computation methods that incorporate the aforementionned process. Algorithmic Information Theory (AIT) appears to be such a framework, as it is consistent with the human perception of complexity [5,6,7]. Our method is as follows: we consider events as being elements stored in what we call *base memory*. To reproduce the language features applicable to events, we use *predicates*, i.e., functions assigning a boolean value to events. For instance, the predicate date (·,1_*year*) is *true* of events that occurred last year. Selecting all events from the memory that satisfy a given predicate corresponds to a *filter* operation. It generates another memory that is a subset of the previous one. The filtering operation can then be repeated, selecting fewer events at each iteration, until a singleton memory is reached. This means that the sequence of predicates could unambiguously *retrieve* the unique remaining event. The description complexity of this event can thus be upper-bounded by the number of bits required to describe the filters used in the retrieval process. Figure 1 illustrates this process for the event: “last year’s hottest day”.

The rest of this article is organized as follows. First, we briefly introduce some relevant notions of Algorithmic Information Theory in Section 2.1. We then expose our contribution in Section 2.2 with a formal definition of memorability. We present an implementation of these definitions with two smart-home examples in Section 3. The results of these experiments are then presented and discussed. We explore other related works in Section 4 and explore possible extensions of our work in Section 5.

## 2. Theoretical Framework

### 2.1. Background

Kolmogorov complexity formally quantifies the amount of information required for the computation of a finite binary string (or any object represented by a finite binary string; although the definition holds for some infinite binary strings (think of the representation of the decimals of π), we restrict ourselves here to finite strings [6,8]. The complexity K(s) of a (finite) binary string *s* is the length in bits L(p) of the shortest program *p*, which, if given as input to a universal Turing Machine *U*, outputs *s*.
(1)$$K_{U}\left(s\right)=\min_{p}\left\{L(p)|U(p)=s\right\}\;.$$

The first notable property of this definition is its universality: while the choice of the Turing machine *U* used for the computations appears in the definition of Equation (Equation 1), all results hold, up to an additional constant, if we change the machine. Consider how any Turing-complete programming language can be turned into any other language, using an interpreter or a compiler program. Since any Turing machine $U^{\prime }$ can be emulated by *U* from a finite program $p_{U}$, we have the following inequality:(2)$$K_{U^{\prime }}\left(s\right)\leq l\left(p_{u}\right)+K_{U}\left(s\right)\;.$$

From this first result, we can then define complexity K(s), based on the choice of a reference Turing machine, such that, for any other machine *U* taken from the set TM of Turing machines,
(3)$$\forall U\in \mathrm{TM},\forall s,|K\left(s\right)-K_{u}\left(s\right)|\leq C_{U}\;,$$
where the additional constant $C_{U}$ does not depend on *s*.

Note that the notion of Kolmogorov complexity involves no requirement on the execution time of programs—only their length in bits matters for the computation of complexity. Though Kolmogorov complexity can be shown to be incomputable [6], it can be approximated with upper bounds by exhibiting a program outputting *s*.

Interestingly, Kolmogorov complexity matches the intuitive notion and perception of complexity from a human standpoint. For instance, the complexity of short binary strings evaluated in [7] shows similar results to human perception of complex strings and patterns. More recently [9] used Kolmogorov complexity to solve analogies and showed results close to human expectations.

The bridge between Algorithmic Information Theory (AIT) and human perception of complexity can be pushed farther thanks to the notions of simplicity and unexpectedness, which are sometimes considered to be of uttermost importance in cognitive science [10]. The author of [5] proposes a formal definition of the unexpectedness U(e) of an event, as the difference between an a priori expected causal complexity $K_{w}\left(e\right)$ and the actual observed complexity K(e).
(4)$$\mathrm{Unex}\left(e\right)=K_{w}\left(e\right)-K\left(e\right)\;.$$

This result stems from the understanding that, while Kolmogorov complexity is ideally computed using a Turing machine, it can be used as a proxy for modeling information processing in the human brain, and thus can be used to define a notion of simplicity or complexity of events. Hence, the term $K_{w}\left(e\right)$, which designates the causal complexity, models the cost of information that a hypothetical World Machine—a Turing Machine modeling the person’s understanding of the world—would require to produce the observed outcome. This can be, for instance, the cost of different parameters in a physical model. As such, this quantity is highly dependent on the knowledge that the human subject has of their surrounding environment.

Definition (Equation 4) allows to model phenomena such as coincidences: imagine that you happen to run into someone in a park. If this person has no particular link to you, the event will be quite trivial: the complexity of describing this person will be equivalent to distinguishing her from the global population, which is also roughly equivalent to the (causal) complexity of describing the circumstances having brought this person to be in that park as the same time as you. On the other hand, if you run into your best friend in a park, as the complexity of describing your best friend is significantly lower, the description complexity K(e) drops while the causal complexity $K_{w}\left(e\right)$ remains unchanged. This is why this latter event appears unexpected. By contrast, if you knew beforehand that your best friend tends to walk in this park, the causal complexity $K_{w}\left(e\right)$ would be significantly lower, hence reducing the surprise.

As the work in [5] suggests a link between unexpectedness and cognitive relevance, we propose to define the memorability of an event in a similar way. However, the proposed definition of unexpectedness, by introducing the hypothetical World Machine, results in an uncomputable metrics. Since we want to use this score in applications, we need a definition that is well-defined and computable in practice. We therefore introduce the memorability M(e) of an event as the absolute difference between its description complexity $K_{d}\left(e\right)$ and its expected description complexity $K_{exp}\left(e\right)$:(5)$$M\left(e\right)=|K_{exp}\left(e\right)-K_{d}\left(e\right)|\;.$$

In the original paper [5], exceptionally complex events are described by considering complexity itself as a way to describe the event: see “the Pisa Tower effect” [11]. Here, contrary to the definition of unexpectedness from Equation (Equation 4), we use an absolute value to account for this phenomenon: an exceptionally complex event is considered as memorable as an exceptionally simple one. In the next section, we define computational approximations for the description complexity $K_{d}$ and the expected complexity $K_{exp}$ of events.

### 2.2. Defining and Retrieving Events

We define *events* as data points augmented with a *label* indicating their nature (temperature event, failure event, addition/removal of a device) and a timestamp of occurrence. Formally:(6)e=(l,t,D),
where *l* is the label, *t* the timestamp and D a vector of properties characterizing *e*: its duration, the maximum temperature reached, the sensor name, its position, etc. Labels can also be considered as classes of events, of which each event is a particular instance.

To model how humans are able to describe events by using qualifiers, we use *predicates*: Boolean functions operating on events and, possibly, additional parameters: $\pi \left(e,a_{1},a_{2},\cdots ,a_{n}\right)\mapsto \left\{O,1\right\}$ is a predicate of arity *n* operating on event *e*. In the rest of this paper, we will prefer the equivalent notation π(e,k)↦{0,1}, where *k* is a binary string encoding the sequence of arguments $a_{1},\cdots ,a_{n}$. Using this notation, the predicate π becomes a boolean function operating on $E\times {\left\{0,1\right\}}^{*}$, where E denotes the set of all events:(7)$$\pi :\left\{\begin{array}{ll} E\times {\left\{0,1\right\}}^{*} & \mapsto \{0,1\}\\ \left(e,k\right) & \mapsto \pi_{k}\left(e\right)\;. \end{array}\right.$$

As an example of predicate, consider π=year and *k* a string encoding the number 1, thus constructing the predicate year(e,1), which indicates whether the event *e* occurred 1 year ago. Another example would be the predicate π=date, which can take as argument k=year::month::day, and date(e,y,m,d) is true if and only if event *e* occurred at the specified date.

As events occur, they are stored in the *base memory* $M_{0}$. As they are not directly comparable, the memory $M_{0}$ can be considered as having the structure of an unordered set. We denote by M the set of all subsets of $M_{0}$. By extension, elements of M, i.e., subsets of $M_{0}$, are also called *memories*.

By applying a given predicate π to all events contained in a memory $M\subseteq M_{0}$, and selecting only events satisfying π, one gets another memory $M_{1}\subseteq M\subseteq M_{0}$. We call this operation a *filter*:(8)$$f_{\pi ,k}:\left\{\begin{array}{cc} M & \mapsto M\\ M & \mapsto \{e\in M|\pi_{k}\left(e\right)\}\;. \end{array}\right.$$

For instance, using the same π=year and k=1 as above, we can build the filter $f_{\pi ,k}=\mathrm{last}\_\mathrm{year}$, which selects all events that occurred last year.

As the output of a filter applied to a memory *M* is another memory object $M^{\prime }\subseteq M$, we can compose filter functions. A sequence of such filters is called a *retrieval path,*
(9)$$p=(f_{\pi_{1},k_{1}},\cdots ,f_{\pi_{n},k_{n}})\;.$$

By definition, $p\left(M\right)=f_{\pi_{n},k_{n}}\left(\cdots \left(f_{\pi_{1},k_{1}}\left(M\right)\right)\right)$. In case the result of the operation p(M) contains a single element *e*, we say that the path *pretrieves* the element *e* from *M*, and write p(M)=e. In the example shown in Figure 1, the three filters $f_{1},f_{2},f_{3}$ form a retrieval path retrieving the event “last year’s hottest day” from the base memory $M_{0}$.

### 2.3. Description Complexity of Events

As presented in Section 2.1, we are interested in computing an approximation of the description complexity of an event *e*. From the above definitions, if there is a path *p* retrieving *e* from the base memory $M_{0}$, i.e., $p(M_{0})=e$, this path provides a possible unambiguous description for *e*. We define the description complexity of *e* as the minimum complexity of a path *p* retrieving *e* from the base memory $M_{0}$.
(10)$$K_{d}\left(e\right)=\min_{p\in P_{\infty }}\left\{L\left(p\right)|p\left(M_{0}\right)=e\right\}\;,$$
where the bit-length L(p) of a retrieval path is defined as the number of bits of a string encoding the path. If we limit ourselves to prefix-free strings encoding predicates and arguments, the total bit length is given by:(11)$$\begin{array}{cc} L(p) & =L(\left(f_{\pi_{1},k_{1}},\cdots ,f_{\pi_{n},k_{n}}\right)) \end{array}$$(12)$$\begin{array}{cc} \; & =L\left(\pi_{1}\right)+L\left(k_{1}\right)+\cdots +L\left(\pi_{n}\right)+L\left(k_{n}\right)\;, \end{array}$$
where $L(\pi_{i})$ and $L(k_{i})$ denote the length, in bits, required to express the predicate’s concept and program, respectively. This length may vary depending of the encoding choice, see Section 3 for an example.

By considering only a finite number of possible predicates π and arguments *k*, and a maximum path length, we can construct a finite set *P* of possible retrieval paths. By limiting the search over this set, we obtain an upper bound of description complexity, and use this upper bound as an approximation:(13)$$K_{d}\left(e\right)\leq \min_{p\in P\wedge p(M_{0})=e}L\left(p\right)=\min_{p\in P\wedge p(M_{0})=e}\sum_{f_{\pi ,k}\in p}L\left(\pi \right)+L\left(k\right)\;.$$

The approximation of description complexity from Equation (Equation 13) allows for a direct implementation, which is shown in Algorithm 1. This algorithm operates iteratively: starting from the base memory $M_{0}$ (line 1), we apply all possible predicate concepts π from a given finite set Π and programs *k* (lines 6–7), up to a given length max_len bits, and apply them: $M^{\prime }=f_{\pi ,k}\left(M\right)$ (line 12). We then store the pairs $\left(M^{\prime },\mathrm{len}\left(\pi ,k\right)\right)$ in an array ${\mathrm{future}}_{\mathrm{explore}}$. At the end of the iteration, the results of the filters become the memories which will be explored during the next iteration(lines 21–23). Each pass thus explores retrieval paths of increasing length. When a singleton memory is reached, the complexity of its unique element is upper-bounded with the length of the corresponding retrieval path (line 14).
**Algorithm 1:** Iterative computation of the approximate complexity
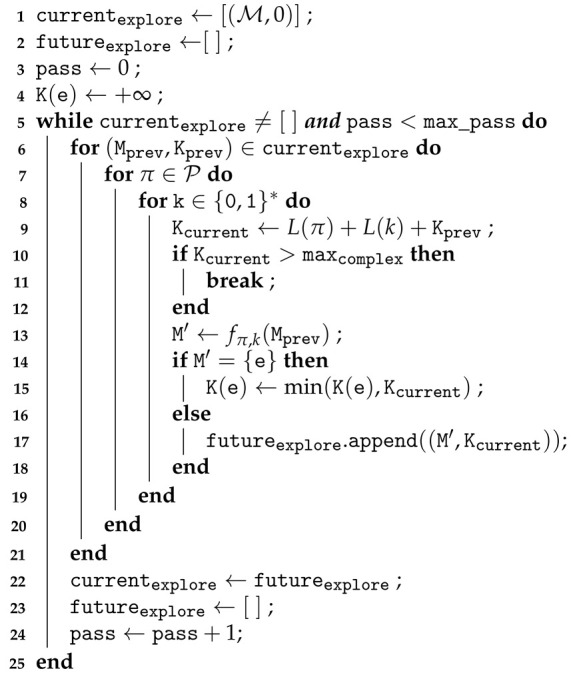


### 2.4. Computing Memorability

As stated in Equation (Equation 5), we define memorability M(e) as the absolute difference between the description complexity of an event and its expected value. As we have just defined $K_{d}\left(e\right)$ and provided an approximation in Equation (Equation 13), we now focus on defining the *expected* description complexity of an event, $K_{exp}\left(e\right)$, which appears in Equation (Equation 5).

$K_{exp}\left(e\right)$ evaluates the complexity that the user, or the system, would expect for the occurrence of event *e* to have, based on their previous knowledge. In our framework, this prior knowledge consists of the base memory $M_{0}$. The expected complexity of the event *e* can be computed with a simple first-order approximation, i.e., estimating the average complexity of “similar events” over the base memory $M_{0}$.

Still, there is a difficulty in defining what should be considered *similar* events. Given that we deal with non-comparable events, we may define the notion of similarity by referring once again to *predicates*. For a given event *e* and a given predicate $\pi_{k}$, we define a $\pi_{k}$-neighborhood of *e* as the set $N_{\pi ,k}\left(e\right)$ of all other events satisfying $\pi_{k}$.
(14)$$N_{\pi ,k}\left(e\right)=\left\{e^{\prime }\in M_{0},\left(e^{\prime }\neq e\right)\wedge \pi_{k}\left(e^{\prime }\right)\right\}\;.$$

Now, when considering, for all possible predicates $\pi_{k}$, the corresponding neighborhoods $N_{\pi ,k}\left(e\right)$, with the convention that $N_{\pi ,k}\left(e\right)=\emptyset$ if *e* does not satisfy $\pi_{k}$, we can compute an average expected complexity for *e*, by summing the complexity of events in all neighborhoods of *e* and dividing the total by the number of events in the neighborhoods:(15)$$K_{exp}\left(e\right)=\frac{\sum_{\pi ,k}\sum_{e^{\prime }\in N_{\pi ,k}\left(e\right)}K_{d}\left(e^{\prime }\right)}{\sum_{\pi ,k}\left|N_{\pi ,k}\left(e\right)\right|}\;.$$

This definition is consistent with the intuitive idea that more similar events should weigh more in the computation. Indeed, if $e^{\prime }$ is very similar to *e*, it will appear in many neighborhoods, since it satisfies most of the predicates that *e* satisfies. Therefore, it will be present in more terms in Equation (Equation 15), and will weigh more in the final result.

This metric solves the different problems exposed in the introduction: by using a universal measure for complexity, bits, it allows to compare values from different dimensions. For instance, it solves the dilemma of recent events: is a big event a long time ago more memorable than a smaller one that occurred only a few minutes ago? With our approach to complexity, each one of these dimensions will scale logarithmically, with the bit length of the required predicate parameters. The balance between them depends on the subjectivity of the system, which is encoded in the intrinsic complexity of predicates for magnitude and dimension.

### 2.5. Defining Relative Memorability for Abduction

Abductive inference builds upon the computation of the memorability score. *Knowing* that we want to find a cause *c* for an observed effect *e*, we try to find the most remarkable event in memory that is related to *e*. While our “memorability” score identifies remarkable past events, it does not take into account their relatedness to *e*.

The knowledge attached to the occurrence of *e* can be integrated into the description complexity definition by using conditional complexity $K_{d}\left(c|e\right)$: The information contained in *e* is considered as given, and therefore as “free” in terms of complexity. For instance, when looking for a cause of an anomaly in the living room, other anomalies occurring in the same living room will be simpler, as the location “living room” is already known from the observation of the current anomaly.

Formally, we now consider that knowledge of the effect is given. This consists, for instance, of appending a description of effect *e* to all programs *k*: $\pi_{e::k}\left(c\right)$, where :: is the *append* operation. The set of paths obtained with such predicates is noted $P_{e}^{\infty }$. This *append* operation is free in terms of bit-length in the computation of complexity, since the effect event *e* is an input of the problem. Therefore, we have $L^{\prime }\left(\pi_{k::e}\right)=L\left(\pi_{k}\right)=L\left(\pi \right)+L\left(k\right)$. We obtain a definition for the conditional description complexity:(16)$$\begin{array}{cc} K_{d}\left(c|e\right) & =\min_{p\in P_{e}^{\infty }}\left\{L^{\prime }\left(p\right),\;p\left(M_{0}\right)=c\right\}\;, \end{array}$$(17)$$\begin{array}{cc} \; & \;=\min_{p\in P_{e}^{\infty }}\left\{\sum_{f_{\pi ,k::e}\in p}L\left(\pi \right)+L\left(k\right),\;p\left(M_{0}\right)=c\right\}\;. \end{array}$$

This new conditional description complexity translates the additional information provided to the system when answering a user’s request. It can then be averaged over similar events to compute the expected conditional description complexity, $K_{exp}\left(c|e\right)$. From this, we come to the definition of the *conditional memorability*, which measures how memorable an event *c* turns out to be in the context of the occurrence of another event *e*:(18)$$M\left(c|e\right)=|K_{exp}\left(c|e\right)-K_{d}\left(c|e\right)|\;.$$

Conditional memorability encapsulates the idea presented as the motivation of this paper: when confronted with a surprising situation, and in the absence of any other source of information, events that appear more memorable than others with regards to the target event will be selected as potential causes. As such, our conditional memorability score provides a ranking that can be used for abductive inference.

## 3. Experiments

To illustrate our approach to memorability, we designed several experiments: the first one is merely a toy example to illustrate the notion of memorability, while the two others are inspired from realistic use cases that may arise in the context of smart homes.

### 3.1. Realistic Examples

We designed two different setups to test our approach. Both are inspired from smart home use cases. This choice of configuration was motivated by the challenges posed by smart homes for abductive inference: (i) as the number of connected devices increases, more events are recorded, making the detection of memorable events more important; (ii) smart homes are prone to experiencing atypical situations, highly dependent on the context, for which pre-established relations might fail to find good abduction candidates. Our choice was also motivated by the existence of previous work [12] involving smart home simulations capable of quickly generating data from which we could extract events and test our methods.

#### 3.1.1. The “TV” Scenario

In this setup, we aim to reproduce the example mentioned in Section 1: installing and using for the first time a brand new smart TV has unpredictable effects on the light of the room where the TV is located. Faced with this situation, history-based approaches fail to identify the right hypothesis as there are no previous data for the new TV. To recreate this situation, we created a set of events covering a period of 100 days, corresponding to the past knowledge of the house. Two kinds of events are recorded: “TV event”, corresponding to TV use (old and new); and luminosity events, describing the luminosity of the room at a given time. Low lights occur at night, and can occasionally occur during daytime (for instance if the blinds are down). On the 100th day, a “TV event” is recorded with a different “device” characteristic: it corresponds to the first usage of the new smart TV. Shortly after, the light dims, which is recorded in a “light” event.

#### 3.1.2. The “Temperature” Scenario

We consider an experimental smart home setup with various sensors, which we simulate over a period of time. The smart home simulation data are then processed to identify some predefined events (such as abnormal temperatures). To perform the simulation, we used the iCasa smart home simulator platform [12], to which we added custom modules. iCasa allows the simulation of autonomic systems that can handle internal communications, the possible insertion of new components at runtime, or the deletion or modification of existing components. We used a basic scenario consisting of a house with four rooms, a single user, and an outdoor zone. All four rooms are equipped with a temperature control system in charge of heaters (see Figure 2).

Based on this, we implemented a scenario spanning over 420 days, and comprising a daily cycle of outdoor weather (temperature and sunlight) fluctuations, as well as user movements. All these daily changes create non-noticeable events, serving as a background for our experiments. To produce outstanding events, we randomly generated about 20 special events, spanning over the whole duration of the simulation. These outstanding events were of the following kinds:Unusual weather: the outdoor conditions are set to unusually high or low temperatures.Heater failures: heaters may break down, making them turn off regardless of the command they receive.Device removal/addition: a device is removed, or another one is added to the system.

Values observed from all devices and zones were sampled throughout the simulation. The resulting data (Figure 3) were then used as a basis for our experiments. We then process the time series data to identify and characterize events. Since the ways events are detected is not the focus of our present work (see Section 4), we perform event detection merely based on threshold comparison, e.g., a temperature event is created if temperature measure are above a given threshold for more than a certain amount of time.

### 3.2. Implementing the Complexity Computation

We implemented the computation of both the description complexity and the memorability score into a Python object called the SurpriseAbductionModule. Apart from the base memory of events $M_{0}$, this module contains a set of predefined predicates Π to characterize events. For instance, for scenario 2, the predicates we used were the following:label(e,k), indicating whether the event *e* has the $k^{th}$ most frequent label (meaning that frequent labels are simpler to express than rare ones). In case some labels have the same frequency, an arbitrary rank is used among them. However, given the unlikeliness of this occurrence, the impact on complexity is insignificant (this case did not occur in our test examples with a few hundreds events)).;rank(e,r,a), indicating whether the event *e* is ranked $r^{th}$ for characteristic *a*, where characteristics are encoded by their frequency (again, common characteristics are the simplest ones);day(e,k), indicating whether the event *e* occurred *k* days ago;month(e,k), indicating whether the event *e* occured *k* months ago;location(e,k), indicating whether the event *e* occurred in zone *k*.

The description length L(π,k) of a predicate $\pi_{k}$ is computed as follows: since the set of predicates is finite and known, $L\left(\pi \right)=\log_{2}\left(|\Pi |\right)$ bits are enough to describe the predicate concept π. This approach assigns equal complexity to all predicate concepts. Though this choice may be questionable when using many concepts, as humans do, we used this simplification, as our examples rely on few predicates. To encode the argument *k* of the predicate, we used the widely used prefix-free Elias delta code [13], which requires $L\left(k\right)=\log_{2}\left(k\right)+2\log_{2}\left(\log_{2}\left(k\right)+1\right)+1]$ bits. The total cost of describing $\pi_{k}$, therefore, is
(19)$$L\left(\pi ,k\right)=\log_{2}\left(|\Pi |\right)+\log_{2}\left(k\right)+2\log_{2}\left(\log_{2}\left(k\right)+1\right)+1\;.$$

With a straightforward implementation of memory, predicates and filters, we could run Algorithm 1. However, it was too time-consuming to be usable in realistic scenarios with hundreds or thousands of events to consider. In order to facilitate and speed up computations, we implemented the following improvements:The memory object was augmented with various built-in rankings, allowing for faster operations during filtering. For instance, since the memory object keeps a mapping from timestamp to events, one can perform a quick filtering by date without having to loop over all stored elements. This convenient mapping, however, is not directly used to retrieve events by their date of occurrence, so as to preserve the theoretical model of memory as an unordered set, as presented in Section 2.2;Each of these predicates holds the property that, in addition to True and False, they can return another value, None, which is theoretically treated as False, but carries the additional information that this predicate concept will also be false for any other element of the memory for any subsequent program *k*. This allows to effectively break the innermost loop in Algorithm 1;Some of the filters, for instance, the date and rank filters, were hard-written. Events can be selected from these precomputed mappings over the memory objects, rather than by testing a predicate over all memory elements.

Our code was written in Python. Examples are presented in the form of Jupyter Notebooks, which allow to quickly reproduce our results and figures. All code is available on our GitHub: https://github.com/EtienneHouze/memorability_code (accessed on 27 January 2022). Figures from the code are interactive: hovering the mouse above points displays the iD of the event, as well as the predicates used in the optimal retrieval path.

### 3.3. Results

#### 3.3.1. The Wedding Day

This first example highlights the notion of memorability: over a lifetime, some days are more memorable than others to a person. This is particularly true for a once-in-a-lifetime event, such as one’s wedding (even though there could be several in one’s life, weddings remain rare events). To illustrate this, we ran the following setup: a thousand events, each one representing one day, were generated. One of them was flagged as being the person’s marriage. We also defined three predicates: marriage(e) is true iff the day corresponds to the person’s marriage, days_ago(e,k) is true if the day occurred *k* days ago, and $\mathrm{days}\_\mathrm{from}(e,e^{\prime },k)$ is true if the event event occurred k days from event $e^{\prime }$. Using these predicates in this experiment produced the results in Figure 4. The high memorability of the wedding stems from its uniqueness: as such, it does not require any additional information to be described. Similarly, the days close to it require small temporal information to be described: “*the day after my wedding*”, hence the memorability score.

Aside from the noticeable peak values, some artifacts are visible: a “valley” is visible on both sides of the wedding peak, and a discontinuity in the derivative is visible around day 700. Both these phenomena originate from the formula from Equation (Equation 5): $M\left(e\right)=|K_{d}\left(e\right)-K_{exp}\left(e\right)|$. Here, in this simple setup, the expected complexity $K_{exp}$ is the same for all days and corresponds to the average complexity of a day. Hence, the discontinuous derivatives that occur around 0 are due to the absolute value operation in the memorability definition. In addition, the “valleys” stem from the days that are close enough to the wedding to be described relative to it, but remain far enough that the complexity gained by this new description is lower than expected. In that sense, these days are even less memorable than average.

This phenomenon illustrates that, while our metric identifies a few outlier events as being memorable, it is not suited to define what are the most ordinary events. This is also visible on the left end of Figure 4: the most complex events are slightly above the average complexity, but not enough to be clearly distinguished. However, the amplitude of this phenomenon remains small relative to the memorability peak surrounding the wedding event. In fact, for similar situations where the complexity of most events is roughly logarithmic (which is the case when events can be retrieved using their order of appearance), for *N* events, the expected complexity is
(20)$$K_{exp}\approx \frac{1}{N}\sum_{k=0}^{N}\log\left(k\right)=\left(1+\frac{1}{2N}\right)\log\left(N\right)+o\left(\frac{\log(N)}{N}\right)\;.$$

The complexity of the most complex event being equal to log(N)+C+o(C), where *C* is a constant, the amplitude of the “left end tail” is bounded and converges towards a finite value *C*.

#### 3.3.2. The “TV” Scenario

##### Memorability

The computation of the description complexity measure for the 2500 recorded events took around 90 seconds using an i7-8565u-equipped laptop. The resulting memorability scores are shown in Figure 5. We can observe that, on average, recent events are given a higher score: this reflects the cost of designating an event by the time elapsed since its occurrence. Furthermore, we can see that some light events, in blue, are more memorable than the main sequence. These events correspond to either events that occurred simultaneously to TV events, in pink: as they are simultaneous to another event, they require additional information to be singled out, temporal information not being enough. Thus, they appear as “surprisingly” more complex than the rest of their kind, hence more memorable.

Some light events also appear more memorable than the rest: they are events when light was surprisingly low given the hour and, therefore, are easier to retrieve. While these general observations are consistent with an a priori intuition, the results are dependent on the choice of predicates used for the computation. See Section 3.4 for a discussion on this dependence.

##### Abduction

To show the potential application to abductive inference, we use our approach for abduction in the first scenario, to see if memorability alone can find the brand new TV to be a reasonable cause for the sudden light dimming.

The results of our algorithm are presented in Table 1: the system correctly identifies the new TV as being the cause. The reason for this choice is that, since the smart TV’s device ID is unique among all other events of type “TV”, its description complexity stands out as being significantly lower than the others, and therefore entails a high memorability score. While our method does not guarantee the correctness of the hypothesis (in fact, abductive reasoning cannot offer such a guarantee [1]), it provides an alternate hypothesis which corresponds to what a human may have suspected in the case where previous knowledge is unavailable.

#### 3.3.3. The “Temperature” Scenario

For this scenario, 578 events were recorded from the setup described in Section 3.1. The computation of the memorability and complexity scores required around 30 s on a commercial laptop equipped with an i7-8565u CPU. Four loops of Algorithm 1 where required (as additional loops did not improve scores).

Similar to the previous scenario, the general trend of events appears as a time-dependent logarithmic score for most events: this corresponds to events for which the simplest retrieval path consists of describing the elapsed duration since their occurrence using the day predicate. As such, it appears that most days are considered “usual” according to our memorability score. This effect produces the main logarithmic sequence of blue dots. On the other hand, some events stand out in terms of complexity: some appear simpler, as they can be distinguished by using their rank along some axis (“the hottest day”, “the second coldest day”), or the rare occurrence of their kind (“the only recorded failure of the heater”).

Event complexity is displayed in Figure 6: similar to what was observed in the wedding day example, a main sequence of events with logarithmic complexity appears: this corresponds to days that can be best described by their time of occurrence. The corresponding “memorability score” is shown in Figure 7. Events from the main sequence are mostly considered non-memorable. On the other hand, some events stand out: for instance, events 20 and 329, which are respectively the hottest and coldest days recorded, or event 183 which correspond to the rare type *device_removal*. Since our memorability measure treats unusually complex or unusually simple events the same way (from the absolute value operation in Equation (Equation 4)), we observe that some events are memorable due to their context only. For instance, the group to which event 149 belongs appears more complex than expected: the same event occurring simultaneously in all four rooms of the house make each instance harder to discern. Table 2 illustrates this by exhibiting the retrieval paths used for complexity computation for these events.

Given that we generated the data used for this experiment, it is possible to flag all perturbation events apart from the usual daily events and evaluate how a detection based on the “memorability” score would perform in distinguishing these events. Even if event detection is not the main purpose of memorability, we conducted the following experiment: we identified 20 events from the “Temperature scenario” that were the result of hand-made perturbation in the smart home simulation (such as days where the outdoor temperature was set to a abnormally high temperature, device removal/addition) as ground-truth events. Then, we used a memorability-based detector (i.e., flagging all events with memorability above a given threshold as “true” events) and tested different threshold values to observe the True-Positive and False-Positive rates. The result of this experiment is presented as a ROC curve in Figure 8 This illustrates the memorability score’s performance as a classifying tool for “out-of-the norm” events. In this example, misclassification has been observed to result from different phenomena: (i) recent events are memorable with our metrics, while not being ground-truth events; (ii) as events are defined on a daily basis, this classification may not be suitable for days-long events (e.g., events 329 and 339 correspond in fact to the same cold night generated in the data), which adds unnecessary information to their description and therefore hinders their memorability score. While the former is a consequence of how our memorability score considers recent events as particularly memorable (this can be understood as a desired feature for such a metrics), the latter stems from event detection and definition, and could be improved by further developments.

### 3.4. Discussion: The Subjectivity of Predicates

For humans, the notion of memorability and event complexity is highly subjective: the same event may appear usual for a person, while being exceptional for another. Our approach to memorability aims to reproduce this subjectivity while providing a formal canvas for memorability computations. Subjectivity is incorporated through the notion of predicates and their complexity, as they reflect the perception a human has of her surrounding environment.

In Figure 9, we present this subjectivity by comparing the analyses of the same base memory of events, generated using the same setup as for Example 1, using three different sets of predicates. The resulting figures show different phenomena based on the predicates available to the module. In Figure 9a, only time and labels were captured by predicates. As a result, the general trend of the curve is logarithmic, as events from *T* days ago require O(L(T))) bits of information. Two main sequences of light events appear: as some light events were recorded simultaneously as TV events, they require the additional information of their label to be retrieved. In Figure 9b, we added predicates qualifying the light intensity, along with a day/night distinction. This added capacity isolated some light events as much simpler: they all appear as aligned green points. These events are times when the light was low during the day, or high during the night. As such, they are outliers, and therefore require less information to be retrieved.

Finally, Figure 9c was created from a module which has the ability to retrieve any event from a memory of size *N*, at the expense of O(L(N)) bits of information. This added capacity has the immediate effect of setting a clear upper limit to the description complexity of items, since any item can be retrieved using L(N) bits (in this example, this limit is around 25 bits). While this kind of direct retrieval is trivial in computer science (one could use the memory address of any stored event), its correlation to human cognition is not obvious: can humans be considered to have this ability to select any event from their entire memory, without restriction regarding their nature, their time of occurrence, their magnitude? However, this highlights an interesting phenomenon: using this direct retrieval, the module makes no distinction, complexity-wise, between events past a given threshold. In other words, there is a generic “uninteresting” category of events, among which the module makes no distinction.

In addition to the number of available predicate concepts, one might also tweak their complexity. In our example scenarios, we used only a handful of predicates, hence we chose to assign all predicate concepts the same bit description length, log(|Π|) (see Equation (Equation 19)). When using more predicate concepts, we may instead give different costs to some of them, to take into account the complexity difference between them: for instance, the generic time concept year would likely require fewer bits than a predicate tun (an old Mayan time unit, corresponding to roughly 360 days). Similar to the selection of predicates used, the complexity assigned to each predicate concept, that is, their encoding, denotes the subjectivity of the user.

## 4. Related Works

Our work is intended to be integrated into larger-scale frameworks to monitor and detect events in complex environments such as smart homes. Smart homes are often regarded as self-organizing systems [14,15]. As such, they present the capacity to adapt to new goals, new components, new environments. A common framework is autonomic computing, which minimizes users’ interventions in the management of the system [15,16].

In situations where more data are available, we could rely on correlation or causal inference from known relations [17,18]. Relations between inference and complexity have already been studied. The case of inference was one of the motivations for R. Solomonoff to introduce his universal algorithmic probability [19] as a tool to reach an idealized inference machine, creating the notion of complexity simultaneously with Kolmogorov. Subsequently, notions of complexity re-emerged in causal inference: in [20] it was found that, when a causal link exists between two random variables A→B, the decomposition of the joint probability is simpler in the direct than the inverse direction: K(P(A))+K(P(B|A))<K(B)+K(A|B).

The relation between complexity, compression and causality was used in [21] to devise the PACK algorithm. It models a dataset by using a family of decision trees where each tree describes how one variable can be expressed given the others. By choosing the model minimizing the total description length (i.e., description of the model and description of the errors), PACK compresses the dataset while finding relations between variables which can be further analyzed. More recently, [22] used Minimum Description Length to determine, given a joint probability distribution over (X,Y), whether *X* causes *Y* or *Y* causes *X*. Their method is based on tree models and it implies that a model respecting the causal relation will be simpler to describe.

Another area in which AIT can provide original approaches is event mining in data streams. The work in [23] provides a good review of modern approaches and techniques in the field. Some previous work can also be noted for having used AIT techniques to qualify and detect events in time series data. For instance, the authors of [24,25] propose weighted permutation entropy as a proxy for complexity measures in time series data, and use it to find relations between different time series. The work in [26] proposes an MDL approach to find the intrinsic dimensions of time series.

The philosophy of our approach can be related to the “Isolation forests” method [27,28]. It evaluates the isolation of data points by constructing random binary tree classifiers. On average, outlier points will require less operations to be singled out. Using the average height of leaves in the tree as a metrics, this approach succeeds in identifying outlier points without having to define a “typical” point. This approach can be understood in terms of complexity: each node of a binary tree classifier needs a fixed amount of information to be described (it must indicates which variable and threshold are used). Thus, nodes that are located higher in the tree need less information to be described. As such, outliers need less information to be singled out. Compared to ours, this method is tailored for data points found in the same metric space. By using predicates as a proxy for complexity computation, our method is more general, as it is agnostic regarding the nature of events.

While all these works advocate for a strong link between complexity and the discovery of causes, they do not extend the notion up to the point we propose in this paper, namely, using the sole complexity as a tool to express the intuitive notion of memorability, and using it for inference.

## 5. Perspectives

The practical application of the theoretical notions of event memorability M(e) and relative memorability M(c|e) requires further developments. We highlight two of them which seem to us, to date, the most challenging.

First, one limitation of the current approach is the requirement of predefined predicate concepts, from which the different filters are constructed. As an extension, we suggest the possibility of defining such predicates dynamically. One may analyze discriminating dimensions of incoming data and create predicates to name these differences, similar to the contrast operations proposed in [29,30]. For instance, the predicate concept hot can be discovered by discriminating a recent hot day along the temperature axis and naming the difference with the prototypical day [31].

Second, execution time is not part of the theoretical view of complexity, it is of prime importance for practical applications, especially when one considers implementation into real-time systems or embedded devices. While the computation we propose appears to be heavy, and possibly heavier as the number of allowed predicates grows, significant time savings can be achieved by trimming the base memory of past events deemed the most “non-memorable”. For instance, one could retain the 100 most memorable events from the past. The difficulty with this approach is that such operations should be performed without interfering with the complexity computations of new elements: by forgetting some past events, even uninteresting ones, one should make sure to keep track of what made the interesting ones, interesting. Investigation of how to do so can pave the way towards practical implementations and dynamic selection of interesting events and help reduce the memory and computation cost of data-driven applications.

The subjectivity of the memorability score was highlighted in Section 3.4. As is, it can be perceived as a strong limitation of the approach, introducing variables in the choice of predicates. However, this subjectivity is, in our opinion, a feature rather than a limitation. As perceived in examples such as the “Pisa Tower” effect [11] or the wedding day scenario, background and individual knowledge are central to the notion of memorability. In this context, learning the user’s background knowledge via preference learning or Natural Language Processing may pave the way towards better integration and improvement of the memorability score.

## 6. Conclusions

We proposed an approach to evaluate event memorability as a difference between the expected description complexity of an event and its actual value. With our definition, something is memorable if its description requires more information than expected or less information than expected. To formalize this notion, we used principles of minimal description exposed in Algorithmic Information Theory. By defining filter operations from predicates and successively applying these filters, we defined formal ways of describing events, whose length can then be measured to evaluate their description length. From this, memorability is defined as the absolute difference between the average complexity of similar events (representing the expected complexity) and the actual description complexity of the event.

We provided an implementation algorithm to compute this measure for events and showed its application in two smart home examples. These scenarios qualitatively illustrate how our measure fares in comparison with the human notion of memorability, and how this measure can be used to propose relevant hypotheses in an abductive inference process without having further knowledge of the system. We discussed the inherent subjectivity of our approach by highlighting the impact of the choice of predicates for complexity computation in a toy example. We consider extending our approach by including online learning of predicates that would make our approach coincide with the subjectivity of the system’s user.

The ability to identify some events as memorable is useful in the current context of computing where connected devices record many events with heterogeneous characteristics, magnitude and types. In this context, our approach of memorability aims to bring a unifying measure to class some events as “memorable”. Two major applications can be considered for this measure. First, it may enable innovative abduction methods that propose memorable events as relevant causes for new and surprising phenomena, mimicking how humans would proceed in such situations. Second, it can be used to filter out insignificant events and potentially discard them, which can be useful if the user requires an overview of events in a complex systems. Computation time of memorability can be high, but in both cases, it can be computed offline and used when needed: for instance, the system can re-evaluate the memorability of all events once per day.

## Figures and Tables

**Figure 1 entropy-24-00346-f001:**
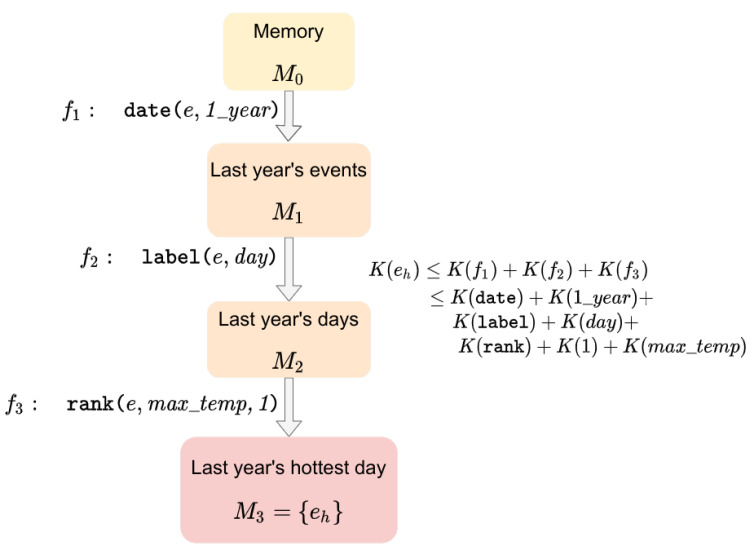
Retrieving an event through successive predicative filters. From the memory (yellow), successive filters select events satisfying the associated predicate (gray arrows). For example, filter $f_{1}$ selects events from last year, i.e., which satisfy the predicate date (event,1_year). In this case, successively applying filters $f_{1}$, $f_{2}$ and $f_{3}$ yields a unique event, $e_{h}$, last year’s hottest day. The complexity of this event can then be upper-bounded by the sum of the complexity of the three filters as they provide an unambiguous way to describe the event within the memory.

**Figure 2 entropy-24-00346-f002:**
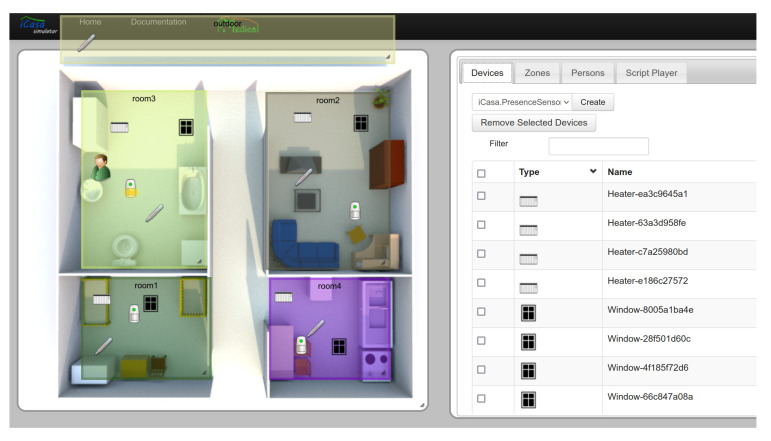
View of the simulator’s web interface provided by iCasa. The four rooms are visible, with their equipment and the user.

**Figure 3 entropy-24-00346-f003:**
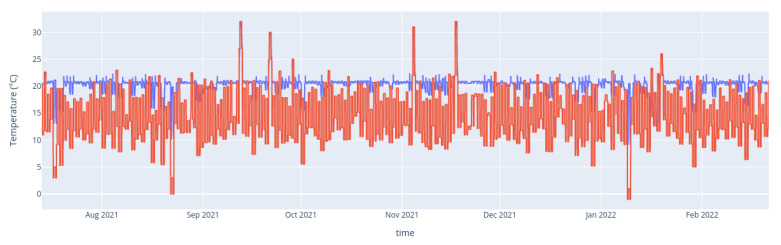
Time series data from the simulation: outdoor temperature (red) and controller temperature of a room (blue). For use in our framework, these time series data were processed by a simple threshold-based event detection.

**Figure 4 entropy-24-00346-f004:**
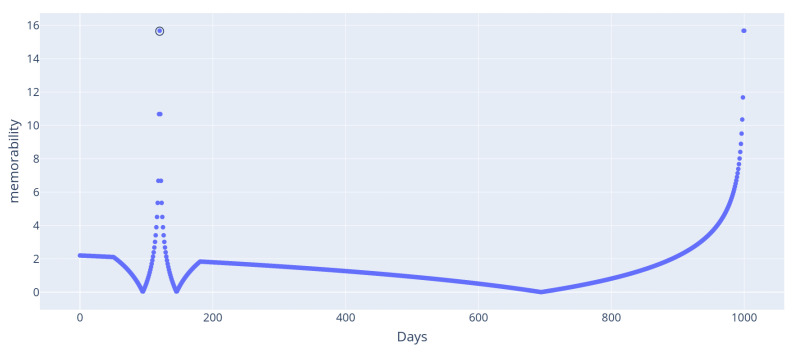
Memorability of days for the marriage example: the wedding day (circled) appears as more memorable, and so do the days directly prior and after it.

**Figure 5 entropy-24-00346-f005:**
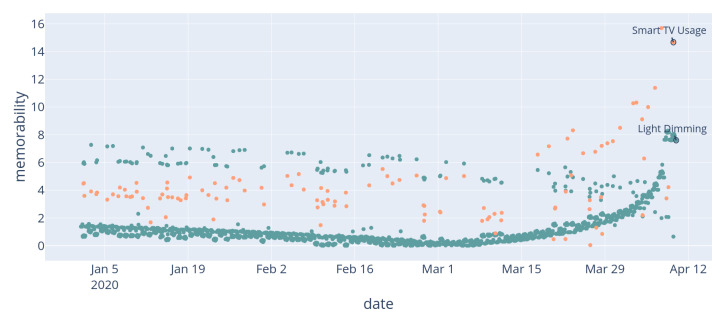
TV scenario. Computed memorability for events. Luminosity events are shown in blue, TV events in pink. The two circled events are the ones mentioned in the scenario.

**Figure 6 entropy-24-00346-f006:**
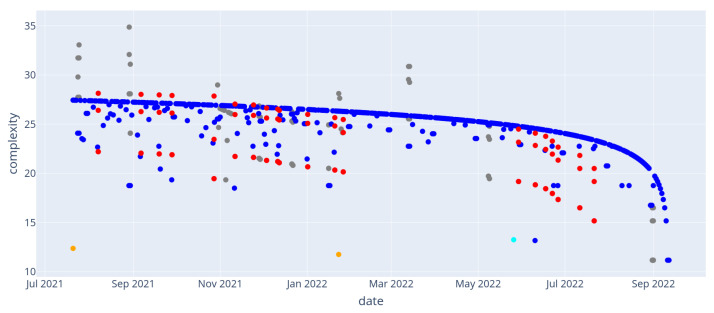
Temperature scenario. Description complexities of events with retrieval paths of length at most 4. Events of type “day” (blue), “hot” (red), “cold” (gray), “device removal” (orange) and “device addition” (cyan) are shown.

**Figure 7 entropy-24-00346-f007:**
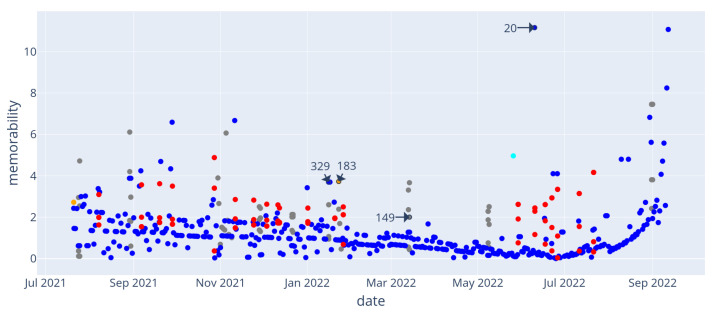
Temperature scenario. Memorability score for events in memory. Events mentioned in the text are highlighted.

**Figure 8 entropy-24-00346-f008:**
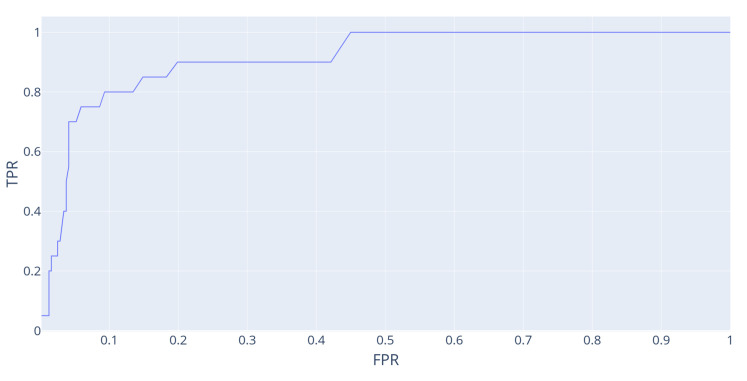
Temperature scenario. Experimental ROC curve (True-Positive Rate against False-Positive Rate) for a classifier which compares the memorability of events to a given threshold, which we vary to change the sensitivity of our detector. Ground-truth events are manually flagged as such as they correspond to the manually generated perturbations of the smart home simulation.

**Figure 9 entropy-24-00346-f009:**
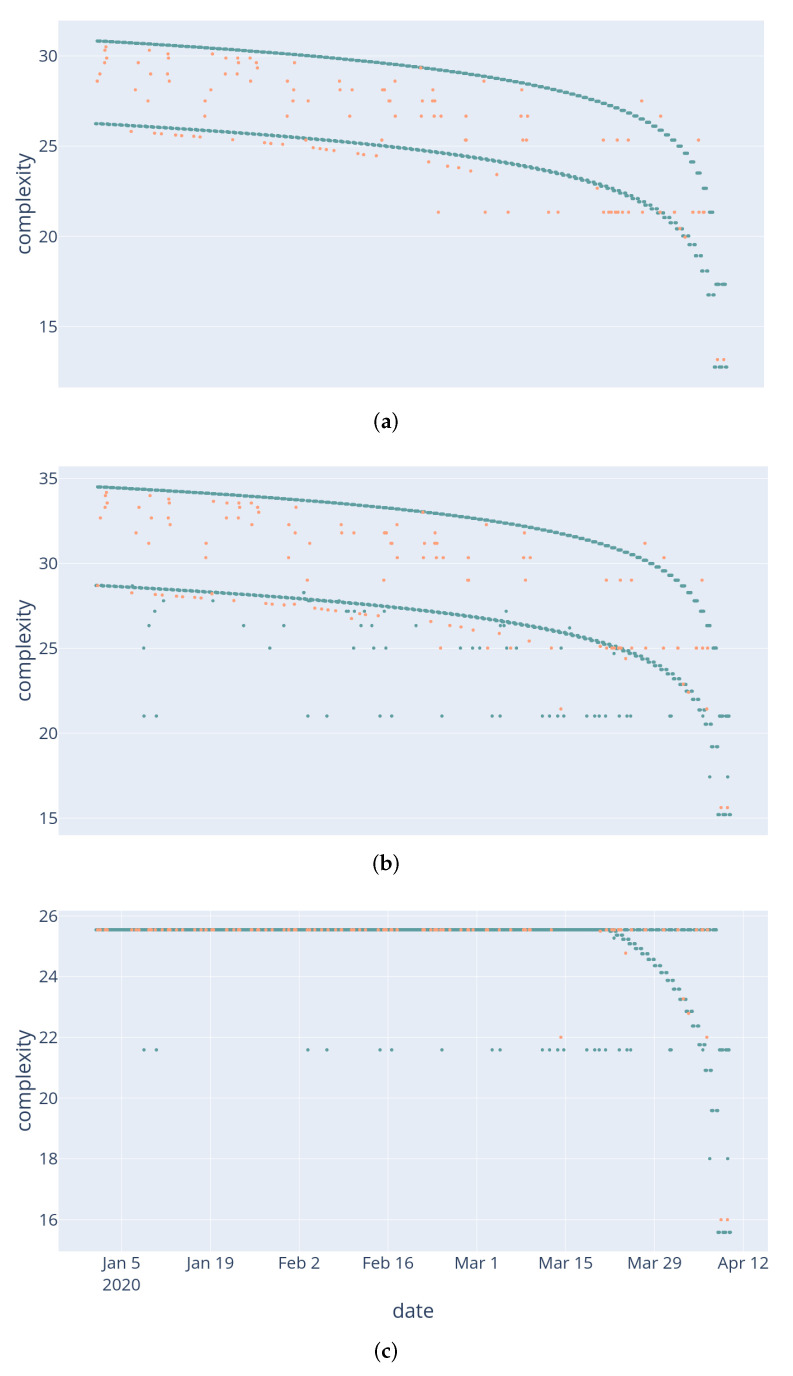
TV scenario, variation. The same memory of events, analyzed using three different sets of predicates. Light events are in blue, TV events—in salmon. (**a**) uses only time-related predicates (days and hours ago), while (**b**) adds label predicates alongside with “dark” and day/night predicates. (**c**) adds the possibility to directly select any event from the memory of size *N*, at the cost of l(N) bits.

**Table 1 entropy-24-00346-t001:** TV Scenario. Output of the memorability-based abduction module: top 3 events for the relative memorability metrics.

Event ID	Description	Relative Memorability (bits)
2513	Use of smart TV	16.76
2427	Last use of the old TV	14.81
2411	Second-last use of the old TV	11.21

**Table 2 entropy-24-00346-t002:** Temperature scenario. Selected events with their shortest retrieval path.

Event iD	Event Type	Retrieval Path
20	day	Label(“day”), AxisRank(0, “max_temp”)
329	day	Label(“day”), AxisRank(0, “min_temp”)
183	deviceRemoval	Label(“deviceRemoval”), Day(0)
149	cold	Label(“cold”), Day(2), Device(“thermo_2”)

## Data Availability

All code and data used for the experiments can be found at https://github.com/EtienneHouze/memorability_code (accessed on 27 January 2022). The iCasa smart home simulator from the Adele research Group, which was used to generate sensor data, can be found at: http://adeleresearchgroup.github.io/iCasa/snapshot/index.html (accessed on the 27 January 2022).

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
