# Peer review of "What Should I Notice? Using Algorithmic Information Theory to Evaluate the Memorability of Events in Smart Homes"

_entropy, 2022, doi:10.3390/e24030346_

Round 1

Reviewer 1 Report

The authors present an interesting idea, that of measuring of memorable an event is. They assume that the most memorable events are likely to require less words to be described and, therefore, they chose the Algorithmic Information Theory framework to build a theory on this subject. The paper is well-written and easy to follow. The ideas and theory are well presented and some experimental results are included.

Memorability has an important subjective component. Something that is memorable for a certain person may be hard to remember to someone else. Hence, I believe that this type of study needs to have a strong support from areas such has psychology and cognitive science, specially when designing the experimental part. In fact, the experimental part of this paper is the one that leaves me somewhat unsatisfied. I understand that IoT in smart homes may be a setup already available to the authors, but I believe the paper would gain a lot with a more comprehensive and diverse experimental setup. Nevertheless, I also believe that the paper, as is, contains already enough interesting material.

Some minor typos:

  • The title, "Alogithmic"
  • Abstract, line 4, "human" -> "humans"
  • Page 5, footnote 2, a ref is missing
  • Page 10, footnote 4, "we we"
  • Line 375, "king"?

Author Response

Dear Reviewer,
We have read your review and taken it into account in this new version of our paper.
The main modifications, aside from the English and grammar mistakes that you pointed out, are the following:
- We have added an "example 0", which is a toy example not from the smart home context: considering one's wedding day and its complexity. This example illustrate the very principle of the memorability score, as it identifies one's wedding, and the surrounding days, as remarkable. It also allows a small discussion on the possible noise affecting less memorable events. - Also, we have added, in the ``Perspective'' section, a paragraph addressing the problem of memorability and stating that this is a desired feature for memorability. - In the conclusion section, we have highlighted the possible use cases for the technique.
We hope that these modifications address most of your comments.
Sincerely,
Etienne Houzé, on behalf of the authors

Reviewer 2 Report

the algorithm for calculating a measurement on events has found its application in two examples of smart homes, relating it to the measurement that a human being can perform. what added value can it bring to a decision-making process and in how much time can it be considered reliable?

Author Response

Dear reviewer,

Thank you for your review of our article. We have made the following changes to the manuscript:

- We have added an "example 0", which is a toy example not from the smart home context: considering one's wedding day and its complexity. This example illustrate the very principle of the memorability score, as it identifies one's wedding, and the surrounding days, as remarkable. It also allows a small discussion on the possible noise affecting less memorable events. - Also, we have added, in the ``Perspective'' section, a paragraph addressing the problem of memorability and stating that this is a desired feature for memorability. - In the conclusion section, we have highlighted the possible use cases for the technique.

We hope that these changes add material to our paper, and that they answer your comments.

Best Regards,

Étienne Houzé, on behalf of the authors